# Cost of cardiovascular diseases and renal complications in people with type 2 diabetes mellitus in the Kingdom of Saudi Arabia: A retrospective analysis of claims database

**Ahmed Hamden Al-Jedai**[1,2]\*, **Hajer Yousef Almudaiheem**[3©], **Dema Abdulrahman Alissa**[3©], **Hadi Saeed Al-Enazy**[4,5,6,7,8,9,10,11©], **Ghazwa B. Korayem**[12©], **Ahlam Alghamdi**[13,14©], **Shabab Alghamdi**[15©]

1 Therapeutic Affairs, Ministry of Health, Riyadh, Saudi Arabia, 2 Clinical Pharmacy, Colleges of Pharmacy and Medicine, Alfaisal University, Riyadh, Saudi Arabia, 3 Therapeutic Affairs Deputyship, Ministry of Health, Riyadh, Saudi Arabia, 4 Department of Family Medicine, Johns Hopkins Aramco Healthcare, Dhahran, Saudi Arabia, 5 Department of Emergency Medicine and Resuscitation, Johns Hopkins Aramco Healthcare, Dhahran, Saudi Arabia, 6 Department of Simulation and Medical Education, Johns Hopkins Aramco Healthcare, Dhahran, Saudi Arabia, 7 Wellness Institute, Johns Hopkins Aramco Healthcare, Dhahran, Saudi Arabia, 8 Central Board for Accreditation of Healthcare Institutions, Riyadh, Saudi Arabia, 9 Council of Health Insurance, Riyadh, Saudi Arabia, 10 Family Medicine Scientific Committee, The Saudi Commission for Health Specialties, Riyadh, Saudi Arabia, 11 Eastern Province Office, Saudi Society of Family Medicine, Riyadh, Saudi Arabia, 12 Department of Pharmacy Practice, College of Pharmacy, Princess Nourah Bint Abdulrahman University, Riyadh, Saudi Arabia, 13 Princess Nourah University, Riyadh, Saudi Arabia, 14 Pharmaceutical Care Services, King Abdullah Bin Abdulaziz University Hospital, Riyadh, Saudi Arabia, 15 Council of Cooperative Health Insurance, Riyadh, Saudi Arabia

© These authors contributed equally to this work.
\* aljedai@hotmail.com

## Abstract

### Background

The burden of macro- and microvascular complications in patients with Type 2 diabetes mellitus (T2DM) is substantial in Middle East countries. The current study assessed the healthcare resource utilization (HCRU) and costs related to cardiovascular and renal complications among patients with T2DM.

### Methodology

This non-interventional, longitudinal, retrospective, cohort study collected secondary data from three insurance claims databases across Kingdom of Saudi Arabia (KSA) of patients diagnosed with T2DM. The study included adult patients aged ≥18 years diagnosed with first cardiovascular disease (CVD) during index time period and at least one T2DM claim anytime during the study time period. The primary analyses were conducted per database, stratified by three cohorts; patients with at least one claim every six months during the 1-year pre-index and 1-year post-index period (cohort 1), patients with at least one claim every six months during the 1-year pre-index, and two years post-index period (cohort 2) and patients with at least one claim every six months during the 1-year pre-index and 3-year

**Data Availability Statement:** All relevant data are within the paper and its Supporting Information files.

**Funding:** The study was funded by Novo Nordisk, KSA.

**Competing interests:** The authors have declared that no competing interests exist.

post-index period (cohort 3). For each Payer database, demographics, CVD subgroups, HCRU, and costs were analysed. Descriptive statistics were used to analyse the data.

## Results

The study sample comprised of 72–78% male and 22–28% female T2DM patients with CVD and renal complications. Patients in the age group of 35–65 years or above contributed to the significant disease burden. Nearly 68 to 80% of T2DM patients developed one CVD event, and 19 to 31% of patients developed multiple CVD events during the follow-up period. For most patients with comorbid CVD and renal disease, the average HCRU cost for post-index periods was higher compared to 1-year pre-index period across the different visit types and activities.

## Conclusion

The study findings elucidates the need for early initiation of therapies that would reduce the long-term cardiovascular and renal outcomes and the associated costs in patients with T2DM.

## 1. Introduction

Diabetes mellitus and its complications constitute a major public health concern, contributing to substantial morbidity and mortality in both developed and developing countries [1].In 2019, it was estimated that 9.3% (463 million) of the population worldwide suffered from the disease, and the numbers are expected to rise to 10.2% (578 million) by 2030 and 10.9% (700 million) by 2045 [1]. This global diabetes explosion is likely to be fueled by an aging population, economic development, and increasing urbanization leading to more sedentary lifestyles and greater consumption of unhealthy foods linked with obesity [2].Type 2 diabetes mellitus (T2DM) accounts for 90% of all cases of diabetes [3]. In the Middle East and North Africa region, it is estimated that approximately 39 million people have diabetes, and this is expected to rise by 72% reaching 67 million in 2045 [4].According to the World Health Organization (WHO), the Kingdom of Saudi Arabia (KSA) has ranked as the country with second-highest prevalence of diabetes in the Middle East and the seventh globally, with diabetes affecting nearly 7 million individuals [5].

Cardiovascular diseases (CVDs) constitute a major group of comorbidities in patients with T2DM [6].The risk of mortality due to CVD is 2–4 fold higher in patients with T2DM than non-T2DM patients [7]. CVDs also lead to significant disability, reduced quality of life, and increased treatment costs for patients with T2DM [6, 8]. In recent years, focus of diabetic pharmacotherapy has been largely on establishing CV safety and CV benefit of all antidiabetic agents [9].Furthermore, the recently published American Diabetes Association recommend using certain glucose-lowering medications with proven CV benefits in patients with T2DM and atherosclerotic cardiovascular disease (ASCVD) or high ASCVD risk [10].

Evidence indicates that CVD has a substantial impact on direct medical costs both at patient and population level [6, 8]. The KSA reported the highest diabetes-related expenditure among the Gulf Cooperation Council (GCC) countries as it spends 21% of its total health expenditure on diabetes care compared to 16% and 19% in other GCC [7]. In 2013, the direct and indirect costs associated with CVD and diabetes reached almost $11 billion in GCC. This cost is estimated to increase to $67.9 billion by 2022, which is equal to one and a half times the healthcare budget of the GCC countries [11].

Chronic kidney disease is another frequent comorbid condition in T2DM patients and is associated with detrimental clinical outcomes and economic effects. Chronic kidney disease in T2DM patients contributes for increased healthcare resource utilization and costs to patients and healthcare system [12]. In a retrospective cohort study, incremental adjusted costs that occurred over follow-up (from baseline) was substantially higher among patients who progressed from baseline stage 0 to stage 4 CKD compared to those who did not progress. Across all stages of CKD, those who progressed to a higher stage of CKD from baseline had follow-up costs that ranged from 2 to 4 times higher than those who did not progress [13].

Healthcare services in KSA are currently provided free of charge to all Saudi citizens and expatriates working in the public sector, primarily through the Ministry of Health (MOH) [14].The MOH is the largest single Payer in the KSA healthcare system, financing over 60% of the services provided. While for others,healthcare insurance paid by their employers covers Saudi citizens and expatriates working in the private sector [15].

Complementary economic data regarding the cost of CVD and renal complications are critical for payers in understanding the financial implications of the change in prescribing recommendations for improving clinical outcomes in patients with T2DM. However, there is paucity in data pertaining to incremental CVD and renal cost in patients with T2DM. The present study aims to assess the initial and follow-up healthcare resource utilization (HCRU), and costs associated with CVD and renal complications, including breakdown per disease type, and compare with the cost for the preceding year among the patients with T2DM.

## 2. Methods

### 2.1 Study design and setting

This was a non-interventional, longitudinal, retrospective, cohort study based on secondary data from 3 insurance claims databases (defined as "Payer 1", "Payer 2", and "Payer 3" databases), which recorded data across KSA of patients diagnosed with T2DM. The index date for each patient was defined as the date on which the first diagnosis of CVD was identified in patients with T2DM during the study time period (between 01 January 2016 through 31 December 2019 for Payer 1; 1 January 2015 to 31 December 2018 for Payer 2; and 01 January 2015 to 31 December 2019 for Payer 3). Patients in each database were further stratified into cohorts and subcohorts, as explained in Fig 1. Insurance claims data in KSA cover both the native Saudi population and expatriate population. Since this was an observational study, there were no treatments involved in this study.International Classification of Disease, Version 10 Clinical Modification (ICD-10-CM) codes was used to identify diagnoses.

### 2.2 Study population

Patients aged 18 years and above with first CVD diagnosis (index date) in the index time period, and who had at least one T2DM claim anytime during the study time period (defined in Study Design and Setting) were included in the analysis. Patients with any records of paediatric consultations, prescriptions, procedures, or services were not considered eligible for the study. Any patient diagnosed with CVD complication before a diagnosis of T2DM or diagnosed with type 1 diabetes mellitus or gestational diabetes during the study period was excluded from the analysis (Table 1).

### 2.3 Baseline variables and outcomes

For each payer databases, the following variables were analysed.

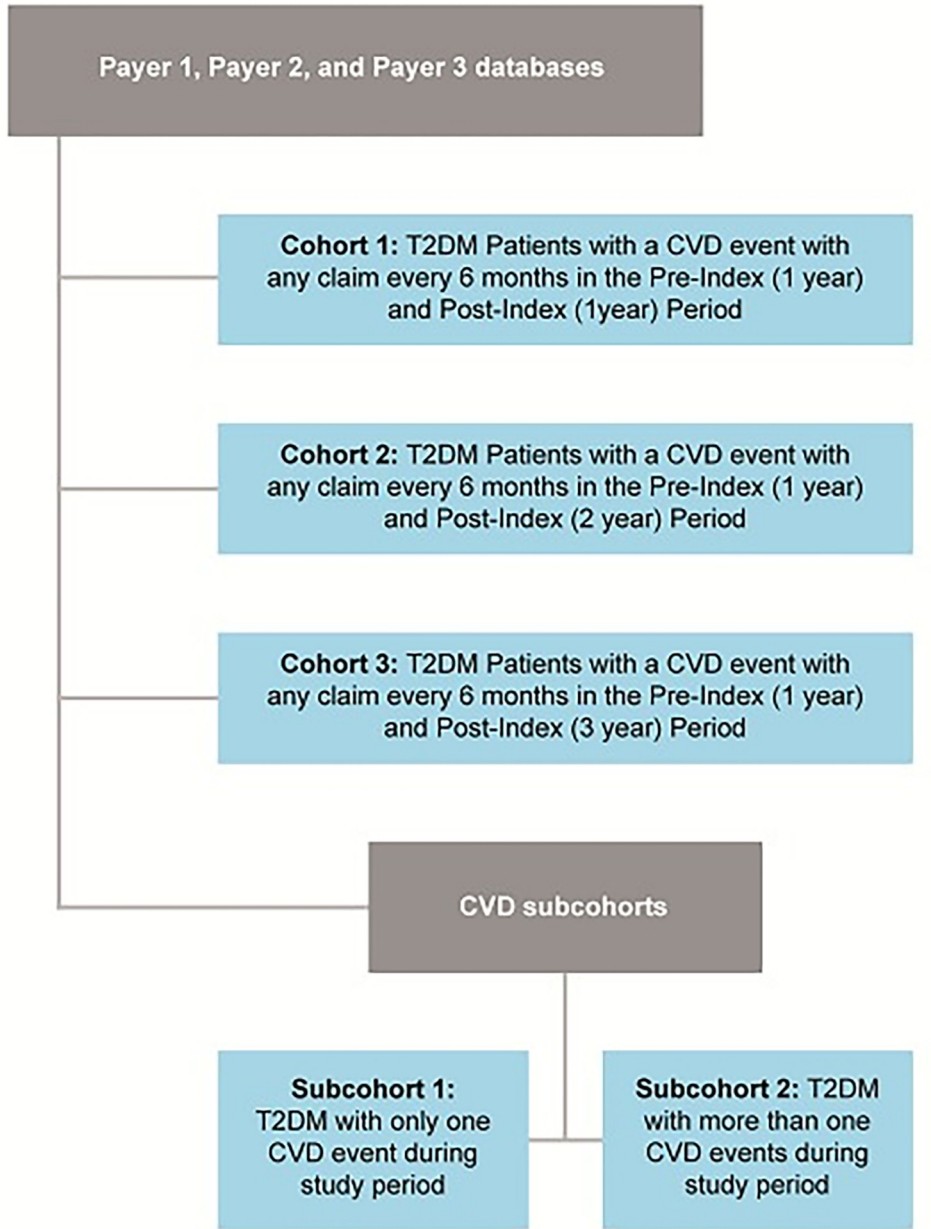

**Fig 1. Patient stratification into cohorts and subcohorts based on follow-up duration and number of CVD events.**

### 2.3.1 Demographics and CVD subgroups

Demographic characteristics such as age (available for Payer 1 and Payer 3), gender (available for Payer 1 and Payer 3), and nationality (available for Payer 1) was extracted from the one year pre-index period claim records.

### 2.3.2. HCRU and costs

Claims and costs (in Saudi Riyal) pertaining to inpatient and outpatient visits and activities including medications, procedures, consultations, services, and consumables were evaluated

**Table 1. Patient disposition.**

| | Payer 1 | Payer 2 | Payer 3 |
|---|---|---|---|
| | Number of patients (%) | Number of patients (%) | Number of patients (%) |
| Patient with first CVD indication on the Index date | 58,493 | 19,636 | 22,964 |
| Patients with at least one T2DM diagnosis in the entire study time period | 29,473 (50) | 7,035 (36) | 22,798 (99) |
| Patients with CVD indication after T2DM | 22,042 (75) | 5,052 (72) | 14,866 (65) |
| Patients with Age ≥18 years on the index date and excluding patient with pediatrics claims | 11,215 (51) | 5,052 (100) | 14,477 (98) |
| Patients without T1DM and gestational diabetes in the entire study time period | 9,419 (84) | 3,748 (74) | 10,910 (75) |

CVD:Cardiovascular disease; T2DM:Type 2 diabetes mellitus;T1DM:Type 1 diabetes mellitus

for post-index period (year 1,2 and 3), across all three databases. All-cause claims included all claims related to T2DM, CVDs, and other comorbid conditions in the patients. Disease-specific claims included claims specific to T2DM and CVD conditions only.

Each database captured overall data from hospitals, pharmacies, clinics, polyclinics, laboratories, and other providers. Databases also included longitudinal, anonymised, and patient-level data, representing a geographically varied range of administrative claims information.

### 2.3.3 Statistical analysis

This study used three analysis sets for Payer 1, Payer 2, and Payer 3. Analyses were conducted individually for each analysis set (data from all analysis sets were not combined). Descriptive statistics were used to summarise patient demographics (i.e., age, gender, and nationality), HCRU, and costs. Descriptive statistics using means for continuous variables and number of observations and percentages for categorical data. Data from each database were extracted and analysed by using SAS version 9.4 software.

### 2.3.4 Ethical consideration

Patient information and consent were not applicable since this study did not involve the interaction or interview with any patient, and the data did not include any individually identifiable variables. Data was anonymous for the purposes of this study and in compliance with applicable laws. Since the core study proposed herein did not involve collecting, using, or transmitting individually identifiable data, the Institutional Review Board'sBoard's approval to conduct this study was not required.

The datasets were secured in an office that meets the requirements of the Health Insurance Portability and Accountability Act of 1996.

## 3. Results

### 3.1 Demographic characteristics

Overall, the data showed that approximately 50% of patients in the study were in 50–64 years age group; >20% of patients were in 35–49 years age group and approximately 20% of patients were in 65–79 years range. A very low proportion of patients (2–5%) were above 80 years of age. Majority of study population was represented by males (>70%). Approximately 25% of patients were native Saudi population and nearly 75% belonged to expatriate population (Table 2).

**Table 2. Demographic characteristics of Payer 1 and Payer 3.**

| Category | | T2DM with one CVD | | | T2DM with multiple CVD | | |
|---|---|---|---|---|---|---|---|
| | | Cohort 1 | Cohort 2 | Cohort 3 | Cohort 1 | Cohort 2 | Cohort 3 |
| Age, years (n) (%) | 18–34 | 96 (3.6) | 43 (4.1) | 9 (4.1) | 12 (1.7) | 7 (2.3) | 1 (1.3) |
| | 35–49 | 685 (25.7) | 283 (26.8) | 48 (22.0) | 149 (21.6) | 70 (23.5) | 20 (26.3) |
| | 50–64 | 1433 (53.7) | 560 (53.1) | 126 (57.8) | 370 (53.5) | 157 (52.7) | 37 (48.7) |
| | 65–79 | 422 (15.8) | 154 (14.6) | 32 (14.7) | 148 (21.4) | 55 (18.5) | 16 (21.1) |
| | 80 and above | 31 (1.2) | 15 (1.4) | 3 (1.4) | 12 (1.7) | 9 (3.0) | 2 (2.6) |
| Gender (n) (%) | Female | 621 (23.3) | 246 (23.3) | 61 (28.0) | 106 (15.3) | 50 (16.8) | 21 (27.6) |
| | Male | 2046 (76.7) | 809 (76.7) | 157 (72.0) | 585 (84.7) | 248 (83.2) | 55 (72.4) |
| Nationality(n) (%) | Saudi | 623 (23.4) | 244 (23.1) | 50 (22.9) | 169 (24.5) | 83 (27.9) | 22 (28.9) |
| | Non-saudi | 2044 (76.6) | 811 (76.9) | 168 (77.1) | 522 (75.5) | 215 (72.1) | 54 (71.1) |
| Age, years (n) (%) | 18–34 | 113 (4.2) | 42 (4.6) | 6 (3.4) | 4 (0.7) | 2 (0.8) | 1 (1.8) |
| | 35–49 | 661 (24.8) | 224 (24.6) | 53 (30.5) | 122 (22.1) | 59 (24.5) | 13 (23.6) |
| | 50–64 | 1317 (49.4) | 472 (51.8) | 88 (50.6) | 275 (49.8) | 118 (49.0) | 30 (54.5) |
| | 65–79 | 496 (18.6) | 145 (15.9) | 24 (13.8) | 130 (23.6) | 54 (22.4) | 11 (20.0) |
| | 80 and above | 81 (3.0) | 28 (3.1) | 3 (1.7) | 21 (3.8) | 8 (3.3) | 0 (0.0) |
| Gender (n) (%) | Female | 732 (27.4) | 261 (28.6) | 39 (22.4) | 119 (21.5) | 49 (20.3) | 12 (21.8) |
| | Male | 1936 (72.6) | 650 (71.4) | 135 (77.6) | 433 (74.8) | 192 (79.7) | 43 (78.2) |

CVD:Cardiovascular disease

(Patient demographics was not captured in Payer 2 database)

Across all payers and cohorts, nearly 68%–80% of T2DM patients had one CVD event and 19%–31% had multiple CVD events, during the index period. The most frequently occurring CVD in the T2DM patients was coronary artery disease (CAD) (40%–80%) followed by stroke and angina (10%–20%).

## 3.2. HCRU claims and cost by the encounter

The average HCRU visits in the inpatient category for both all-cause and disease cause for all the three payers ranged from 1 to 2 visits (Table 3, S1–S5 Tables). A proportional increase in inpatient claims and cost was noted during 1-year and 2-year follow-up period compared to pre-index 1-year period, for both single and multiple CVD events in T2DM patients (Fig 2, Table 3). The increase in inpatient claims and cost was more significant for disease-specific claims as compared to all-cause claims (S3–S5 Tables). In T2DM patients with CRF, inpatient claims and cost was high for both all-cause and disease-specific HCRU. In T2DM patients with multiple CVD events, inpatient claims and cost was high in patients with coprevalent CAD and CRF(Fig 3). Outpatient utilization and costs for both single and multiple CV events and for CRF was higher during 1-year follow-up period compared to pre-index 1-year period (S3–S5 Tables).

## 3.3 HCRU and cost by activity

Medication and consultation costs dominated the visit claims while costs for claims related to medication and procedure were highest for T2DM patients with both single and multiple CVD events, across all payers. Services costs was significantly higher during post-index period in T2DM patients with comorbidities such as CRF and CAD (Table 4; S6–S23 Tables).

**Table 3. Comparison of inpatient and out-patient pre-index and post-index all-cause cost (Payer 1).**

| All-cause | Cohort 1 | | | | | | Cohort 2 | | | | | | | | | Cohort 3 | | | | | | | | | | | |
|---|---|---|---|---|---|---|---|---|---|---|---|---|---|---|---|---|---|---|---|---|---|---|---|---|---|---|---|
| | Pre-Index 1 Yr | | | Post-Index 1 Yr | | | Pre-Index 1 Yr | | | Post-Index 1 Yr | | | Post-Index 2 Yr | | | Pre-Index 1 Yr | | | Post-Index 1 Yr | | | Post-Index 2 Yr | | | Post-Index 3 Yr | | |
| | N | HCRU | Cost | N | HCRU | Cost | N | HCRU | Cost | N | HCRU | Cost | N | HCRU | Cost | N | HCRU | Cost | N | HCRU | Cost | N | HCRU | Cost | N | HCRU | Cost |
| **Inpatient** | | | | | | | | | | | | | | | | | | | | | | | | | | | |
| **T2DM with one CVD $212,754** | | | | | | | | | | | | | | | | | | | | | | | | | | | |
| T2DM+CAD | 201 | 2 | 21,045 | 279 | 1 | 22,988 | 84 | 2 | 17,620 | 106 | 1 | 22,114 | 97 | 2 | 22,120 | 22 | 2 | 19,549 | 26 | 1 | 23,104 | 23 | 2 | 28,772 | 23 | 3 | 41,873 |
| T2DM+Stroke or TIA | 40 | 1 | 18,792 | 71 | 2 | 25,321 | 15 | 1 | 24,412 | 19 | 1 | 32,611 | 13 | 3 | 98,217 | 1 | 1 | 8,544 | 4 | 1 | 15,174 | 3 | 2 | 50,360 | 3 | 1 | 11,845 |
| T2DM+Angina | 15 | 1 | 8,348 | 23 | 1 | 10,841 | 3 | 2 | 9,925 | 3 | 1 | 6,445 | 4 | 1 | 7,452 | | | | | | | | | | | | |
| Others* | 60 | 10 | 113,653 | 80 | 12 | 153,604 | 22 | 6 | 72,835 | 20 | 5 | 75,817 | 26 | 8 | 111,275 | 26 | 5 | 58,855 | 30 | 4 | 47,830 | 24 | 4 | 48,451 | 24 | 4 | 42,043 |
| **T2DM with multiple CVD $399,185** | | | | | | | | | | | | | | | | | | | | | | | | | | | |
| T2DM+ CAD+ Angina | 15 | 1 | 22,944 | 73 | 2 | 32,549 | 7 | 1 | 33,018 | 24 | 2 | 31,437 | 14 | 2 | 25,885 | 4 | 2 | | 2 | 2 | 22,027 | 2 | 2 | 48,618 | | | |
| T2DM+MI+ CAD | 12 | 1 | 19,636 | 23 | 2 | 540,92 | 5 | 1 | 17,985 | 25 | 1 | 29,392 | 12 | 2 | 14,614 | 6 | 1 | | 4 | 1 | 33,389 | 3 | 2 | 13,678 | 3 | 1 | 15,560 |
| T2DM+Stroke or TIA + CAD | 17 | 2 | 23,905 | 49 | 2 | 29,709 | 8 | 2 | 26,391 | 25 | 2 | 24,851 | 20 | 2 | 42,002 | 4 | 3 | 20,963 | 5 | 3 | 31,673 | 6 | 1 | 22,852 | 6 | 2 | 15,558 |
| T2DM + Heart failure + CAD | 16 | 1 | 14,599 | 72 | 1 | 41,745 | 5 | 1 | 24,657 | 8 | 3 | 37,892 | 8 | 2 | 37,666 | 2 | 1 | 5,136 | 1 | 1 | 4,408 | 1 | 1 | 13,030 | 1 | 1 | 4,635 |
| **Out patient** | | | | | | | | | | | | | | | | | | | | | | | | | | | |
| **T2DM with one CVD $119,033** | | | | | | | | | | | | | | | | | | | | | | | | | | | |
| T2DM+CAD | 1,802 | 15 | 9,997 | 1,802 | 16 | 10,609 | 775 | 15 | 10,385 | 775 | 16 | 10,959 | 775 | 17 | 12,011 | 178 | 15 | 10,824 | 178 | 16 | 10,883 | 178 | 19 | 13,547 | 178 | 15 | 10,311 |
| T2DM+Stroke or TIA | 297 | 16 | 9,252 | 297 | 18 | 10,475 | 105 | 14 | 8,537 | 105 | 15 | 8,936 | 105 | 16 | 10,829 | 17 | 14 | 7,898 | 17 | 12 | 6,889 | 17 | 20 | 10,175 | 17 | 16 | 8,425 |
| T2DM+Angina | 219 | 15 | 7,669 | 219 | 14 | 7,075 | 75 | 17 | 6,452 | 75 | 17 | 7,284 | 75 | 17 | 6,601 | 8 | 19 | 5,178 | 8 | 16 | 5,541 | 8 | 20 | 6,172 | 8 | 16 | 5,598 |
| Others | 349 | 147 | 83,831 | 349 | 150 | 90,874 | 100 | 117 | 77,836 | 100 | 115 | 74,785 | 100 | 113 | 88,461 | 15 | 66 | 42,035 | 15 | 65 | 49,149 | 15 | 82 | 74,248 | 15 | 62 | 48,230 |
| **T2DM with multiple CVD$** | | | | | | | | | | | | | | | | | | | | | | | | | | | |
| T2DM+ CAD+ Angina | 160 | 15 | 7,876 | 160 | 17 | 10,756 | 67 | 15 | 8,342 | 67 | 17 | 11,219 | 67 | 19 | 11,720 | 16 | 18 | 9,074 | 16 | 20 | 11,301 | 16 | 22 | 13,137 | 16 | 15 | 10,013 |
| T2DM + MI + CAD | 159 | 12 | 5,867 | 159 | 17 | 10,946 | 57 | 12 | 6,596 | 57 | 17 | 13,029 | 57 | 17 | 11,206 | 10 | 15 | 8,167 | 10 | 21 | 14,302 | 10 | 25 | 16,231 | 10 | 20 | 13,951 |
| T2DM+Stroke or TIA + CAD | 141 | 16 | 11,335 | 141 | 22 | 15,741 | 71 | 15 | 11,654 | 71 | 19 | 14,703 | 71 | 20 | 15,662 | 23 | 17 | 10,820 | 23 | 22 | 15,857 | 23 | 24 | 18,395 | 23 | 20 | 14,752 |
| T2DM + Heart failure + CAD | 67 | 15 | 10,937 | 67 | 20 | 19,354 | 33 | 15 | 9,633 | 33 | 20 | 13,491 | 33 | 17 | 11,720 | 6 | 12 | 11,464 | 6 | 17 | 19,446 | 6 | 15 | 14,937 | 6 | 12 | 14,484 |

CAD = Coronary artery diseases, CVD = Cardiovascular disease, HCRU = Healthcare cost utilization, N = Number of patients, T2DM = Type 2 diabetes mellitus, TIA = Transient ischemic attack

Others* - Atrial fibrillation, cardiac ischemia, Chronic renal failure, Coronary Arterial Revascularization, Dysrhythmia, Heart Failure, Myocardial infarction, Other Cardiovascular Disease, Periphery vascular disease

$—Only the most prevalent Multiple CVD complications of T2DM are included

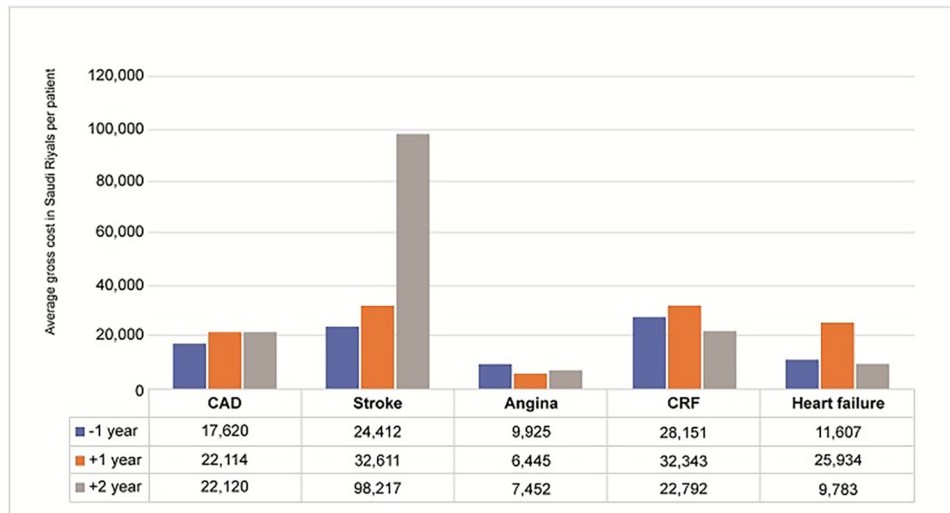

**Fig 2. HCRU costs for inpatient claims during 2-year follow-up period for single CVD and renal events (Payer 1, Cohort 2).**

## 4. Discussion

The alarming rise in the burden of T2DM and associated complications in Middle East countries is a cause of concern. There is limited data in the literature regarding the initial and follow-up HCRU costs associated with complications from CVD and renal disease in T2DM patients in these countries. To our knowledge, this real-world study is the first of its kind to evaluate the initial and follow-up HCRU costs associated with CVD and renal complications, in patients with T2DM in the Saudi population.

This study showed that patients with T2DM in the age group of 50–64 years had an increased incidence of CVD complications, in line with published literature [16, 17]. The

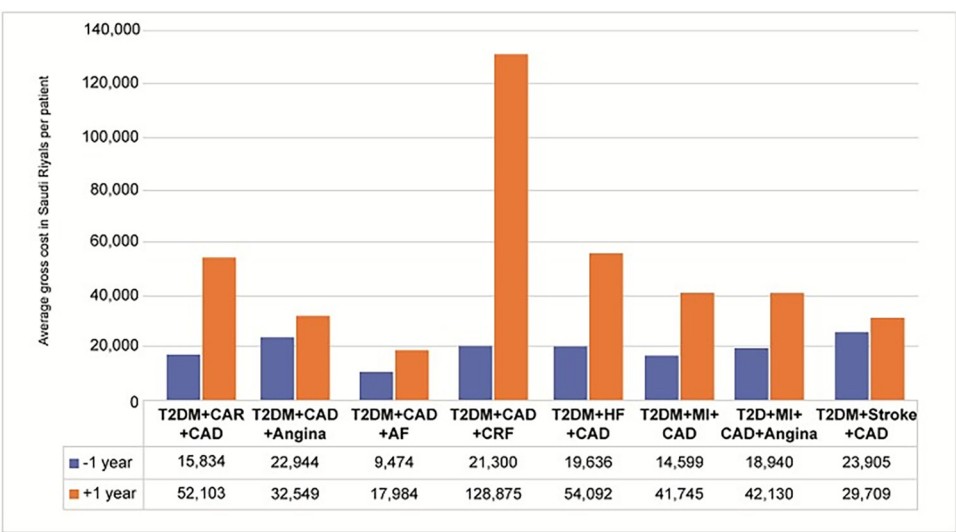

**Fig 3. HCRU cost for inpatient claims for multiple CV and renal events during 1-year follow-up period (Payer 1, Cohort 1).**

**Table 4. HCRU disease-specific claims and costs for various activities during 2-year follow-up period (Payer 3, Cohort 2).**

| Sum of Patient_Count | | | | | | | | | |
|---|---|---|---|---|---|---|---|---|---|
| | Pre-Index 1 Yr | | | Post-Index 1 Yr | | | Post-Index 2 Yr | | |
| Payer 3 | Disease Cause | | | Disease Cause | | | Disease Cause | | |
| Cohort 2 | N | HCRU | Cost | N | HCRU | Cost | N | HCRU | Cost |
| **T2DM With One CVD** | | | | | | | | | |
| **T2DM+Angina** | | | | | | | | | |
| Medication | 138 | 4 | 3,287 | 134 | 5 | 5,073 | 134 | 4 | 3,375 |
| Procedure | 130 | 3 | 2,814 | 130 | 4 | 4,280 | 126 | 3 | 2,011 |
| Consultation | 147 | 4 | 637 | 138 | 5 | 961 | 134 | 4 | 557 |
| Consumables | 13 | 3 | 367 | 29 | 3 | 376 | 19 | 2 | 244 |
| Services | 9 | 1 | 1,744 | 19 | 1 | 1,816 | 4 | 1 | 71 |
| **T2DM+Chronic renal failure1** | | | | | | | | | |
| Medication | 64 | 4 | 5,456 | 64 | 6 | 8,192 | 67 | 6 | 5,258 |
| Procedure | 63 | 3 | 3,659 | 65 | 5 | 8,948 | 66 | 5 | 7,171 |
| Consultation | 67 | 4 | 962 | 66 | 6 | 1,648 | 70 | 5 | 1,104 |
| Consumables | 12 | 3 | 1,247 | 15 | 3 | 1,081 | 17 | 2 | 370 |
| Services | 7 | 1 | 1,388 | 16 | 2 | 12,824 | 12 | 3 | 5,038 |
| Others | | | | 3 | 1 | 43 | 6 | 1 | 19 |
| **T2DM+Coronary Artery Disease** | | | | | | | | | |
| Medication | 347 | 4 | 3,656 | 341 | 6 | 5,651 | 333 | 5 | 3,829 |
| Procedure | 276 | 3 | 2,644 | 290 | 4 | 5,159 | 270 | 3 | 2,416 |
| Consultation | 347 | 4 | 642 | 343 | 6 | 927 | 329 | 5 | 563 |
| Consumables | 45 | 3 | 850 | 56 | 3 | 447 | 47 | 2 | 291 |
| **T2DM+Stroke or TIA** | | | | | | | | | |
| Medication | 148 | 5 | 4,227 | 155 | 6 | 6,020 | 145 | 5 | 3,966 |
| Procedure | 142 | 3 | 3,064 | 140 | 4 | 5,116 | 139 | 3 | 2,877 |
| Consultation | 158 | 5 | 876 | 155 | 6 | 1,607 | 150 | 5 | 839 |
| Consumables | 24 | 3 | 339 | 28 | 3 | 522 | 33 | 3 | 368 |
| Services | 17 | 2 | 4,950 | 21 | 2 | 2,409 | 21 | 1 | 2,681 |
| Others | 9 | 1 | 34 | 7 | 1 | 20 | 5 | 1 | 20 |
| **T2DM With Multiple CVD** | | | | | | | | | |
| Coronary Arterial Revascularization+T2DM+Coronary Artery Disease | | | | | | | | | |
| Medication | 21 | 5 | 6,052 | 23 | 7 | 7,628 | 22 | 7 | 6,326 |
| Procedure | 20 | 4 | 11,133 | 23 | 4 | 9,305 | 21 | 4 | 11,968 |
| Consultation | 23 | 5 | 902 | 23 | 7 | 1,442 | 22 | 7 | 1,337 |
| Consumables | 3 | 2 | 5,434 | 2 | 2 | 3,917 | 1 | 2 | 318 |
| Services | 4 | 1 | 1,424 | 7 | 2 | 2,855 | 6 | 2 | 1,250 |
| Others | 5 | 1 | 63 | 2 | 1 | 11 | 2 | 1 | 0 |
| Coronary Arterial Revascularization+T2DM+Coronary Artery Disease+ Angina | | | | | | | | | |
| Medication | 8 | 4 | 6,437 | 8 | 7 | 10,310 | 8 | 8 | 8,824 |
| Procedure | 8 | 3 | 4,716 | 6 | 5 | 17,844 | 5 | 4 | 12,887 |
| Consultation | 8 | 5 | 998 | 8 | 7 | 1,166 | 8 | 6 | 1,307 |
| Consumables | | | | 2 | 1 | 101 | 1 | 4 | 588 |
| Services | 1 | 1 | 315 | 1 | 2 | 1,500 | 1 | 3 | 9,656 |
| Others | | | | 1 | 2 | 10,800 | 1 | 1 | 0 |
| T2DM+Coronary Artery Disease+ Angina | | | | | | | | | |
| Medication | 72 | 5 | 4,404 | 75 | 8 | 7,611 | 72 | 6 | 4,702 |

(*Continued*)

**Table 4.** (Continued)

| Sum of Patient_Count | | | | | | | | | |
|---|---|---|---|---|---|---|---|---|---|
| | Pre-Index 1 Yr | | | Post-Index 1 Yr | | | Post-Index 2 Yr | | |
| **Payer 3** | **Disease Cause** | | | **Disease Cause** | | | **Disease Cause** | | |
| **Cohort 2** | **N** | **HCRU** | **Cost** | **N** | **HCRU** | **Cost** | **N** | **HCRU** | **Cost** |
| Procedure | 58 | 4 | 2,773 | 67 | 5 | 15,304 | 64 | 4 | 7,410 |
| Consultation | 71 | 5 | 648 | 74 | 8 | 1,347 | 72 | 6 | 792 |
| Consumables | 7 | 2 | 365 | 12 | 3 | 4,879 | 12 | 3 | 981 |
| Services | 12 | 2 | 1,214 | 33 | 2 | 5,372 | 17 | 1 | 2,353 |
| Others | 3 | 1 | 12 | 10 | 1 | 1,811 | 5 | 2 | 216 |
| T2DM+Coronary Artery Disease+ Chronic renal failure | | | | | | | | | |
| Medication | 13 | 4 | 4,326 | 14 | 7 | 9,129 | 13 | 5 | 7,474 |
| Procedure | 12 | 3 | 4,145 | 12 | 5 | 6,635 | 12 | 4 | 7,362 |
| Consultation | 13 | 4 | 790 | 14 | 6 | 2,298 | 13 | 5 | 1,120 |
| Consumables | 1 | 1 | 225 | 2 | 2 | 414 | 5 | 1 | 136 |
| Services | 3 | 1 | 3,395 | 5 | 2 | 9,217 | 7 | 2 | 7,447 |

T2DM: Type 2 diabetes mellitus

incidence of CVD complications was higher among males (72–78%) as compared to females (22–28%), as observed in previous studies [6].Similar to earlier evaluations, study findings indicate that majority of T2DM patients with CRF belonged to age group 50–64 and nearly 76–78% were males [18, 19]. Since this was a private paid insurance, expatriates contributed to 76% of claims across all 3 payer databases studied.

Nearly 68–80% of patients with T2DM had at least one CVD event and CRF during the study period, and multiple CVD complications were noted in 19-31% of patients across all three payers. In a cross-sectional observation study conducted in Gulf, the most prevalent complications associated with T2DM was CKD (44.3%) and CVD (17.3%) and were found to increase substantially with age [20]. In a retrospective study, 97.5% of T2DM patients had atleast one comorbid condition and 88.5% had atleast two; CKD was reported in 24.1% patients and CVD in 21.6% patients and this study also noted an increasing trend in prevalence of comorbidities with advancing age [21]. From these findings we can infer that burden of disease due to comorbid conditions increases with age and progression of disease.

Looking into the HCRU and cost burden, we observed that overall, in majority of T2DM patients with CVD, there was a gradual increase in average HCRU and cost over the 3-year post-index period compared with the 1-year pre-index period. A significant increase in the inpatient numbers and visits for both all-cause and disease-specific causes was noted across all payers during the follow-up period. Observations from a retrospective claims database analysis indicate that higher medical costs for cardiovascular treatment was incurred during initial hospitalisation in patients with T2DM as compared to their non-diabetic counterparts [21]. Furthermore, several reports have demonstrated that CVD events have significant impact on the total and diabetes-related healthcare costs [22–25].

Current study also noted significant cost burden due to comorbid CRF in T2DM patients. Reports from a large US administrative claims database indicate that T2DM patients with newly recognised chronic kidney disease (CKD) had a high prevalence of cardiovascular comorbidities and incurred substantial HCRU and cost. The study emphasised on effective interventions for slowing the progression of CKD [26]. In yet another retrospective study, which estimated incremental all-cause HCRU and costs between T2DM patients who

experienced CVD and renal complications vs. T2DM patients without any complications, significant disease and cost burden was reported in T2DM patients with complications [27]. Therefore payers, providers, and policy makers should collaboratively focus on disease management strategies and interventions that aim at reducing comorbidity-related hospitalisations and thereby healthcare costs, in T2DM patients with complications.

Of note, this is the largest study of its kind in the Middle East that assessed the cost of CVDs and renal complications in patients with T2DM. Nonetheless, certain limitations in the study may have led to bias while interpreting the results. Even though the study investigators tried to reduce the bias in the data, biases related to data duplication, possible misclassification, residual confounding bias, and selection bias may still exist. Another limitation is the retrospective nature of the study. It is worth mentioning that the claims data mostly reflected the expatriate population since all Saudi citizens are provided free healthcare mainly covered by the MOH.

## 5. Conclusion

The current analyses shed light on the substantial economic burden in patients with T2DM with comorbid CVD and renal diseases. The study observed a greater frequency of hospitalization in patients with T2DM with associated CVD and renal comorbidities. The findings also suggest a gradual increase in average healthcare resource cost over the 1-year post-index period compared with the 1-year pre-index period.

Evidence suggests that glucagon-like peptide-1 (GLP-1) agonists are associated with reduced risk of CVS and renal events and sodium-glucose-cotransporter-2 (SGLT2) have improved renal outcomes in patients with T2DM. Therefore, early initiation of these T2DM therapies would reduce long-term cardiovascular and renal outcomes and reduce the associated costs in patients with T2DM.

## Supporting information

**S1 Table. Comparison of inpatient and outpatient pre-index and post-index all-cause cost (Payer 2).**
(DOCX)

**S2 Table. Comparison of inpatient and outpatient pre-index and post-index all-cause cost (Payer 3).**
(DOCX)

**S3 Table. Comparison of inpatient and outpatient pre-index and post-index disease-specific cause cost (Payer 1).**
(DOCX)

**S4 Table. Comparison of inpatient and outpatient pre-index and post-index disease-specific cost (Payer 2).**
(DOCX)

**S5 Table. Comparison of inpatient and outpatient pre-index and post-index disease-specific cause cost (Payer 3).**
(DOCX)

**S6 Table. Comparison of pre-index and post-index all-cause cost for various activities (Payer 1, Cohort 1).**
(DOCX)

**S7 Table. Comparison of pre-index and post-index all-cause cost for various activities (Payer 1, Cohort 2).**
(DOCX)

**S8 Table. Comparison of pre-index and post-index all-cause cost for various activities (Payer 1, Cohort 3).**
(DOCX)

**S9 Table. Comparison of pre-Index and post-index all-cause cost for various activities (Payer 2, Cohort 1).**
(DOCX)

**S10 Table. Comparison of pre-index and post-index all-cause cost for various activities (Payer 2, Cohort 2).**
(DOCX)

**S11 Table. Comparison of pre-index and post-index all-cause cost for various activities (Payer 2, Cohort 3).**
(DOCX)

**S12 Table. Comparison of pre-index and post-index all-cause cost for various activities (Payer 3, Cohort 1).**
(DOCX)

**S13 Table. Comparison of pre-index and post-index all-cause cost for various activities (Payer 3, Cohort 2).**
(DOCX)

**S14 Table. Comparison of pre-index and post-index all-cause cost for various activities (Payer 3, Cohort 3).**
(DOCX)

**S15 Table. Comparison of pre-index and post-index disease-specific cause cost for various activities (Payer 1, Cohort 1).**
(DOCX)

**S16 Table. Comparison of pre-index and post-index disease-specific cause cost for various activities (Payer 1, Cohort 2).**
(DOCX)

**S17 Table. Comparison of pre-index and post-index disease-specific cause cost for various activities (Payer 1, Cohort 3).**
(DOCX)

**S18 Table. Comparison of pre-index and post-index disease-specific cause cost for various activities (Payer 2, Cohort 1).**
(DOCX)

**S19 Table. Comparison of pre-index and post-index disease-specific cause cost for various activities (Payer 2, Cohort 2).**
(DOCX)

**S20 Table. Comparison of pre-index and post-index disease-specific cause cost for various activities (Payer 2, Cohort 3).**
(DOCX)

**S21 Table. Comparison of pre-index and post-index disease-specific cause cost for various activities (Payer 3, Cohort 1).**
(DOCX)

**S22 Table. Comparison of pre-index and post-index disease-specific cause cost for various activities (Payer 3, Cohort 2).**
(DOCX)

**S23 Table. Comparison of pre-index and post-index disease-specific cause cost for various activities (Payer 3, Cohort 3).**
(DOCX)

## Acknowledgments

Concept, design, analysis and medical writing was supported by Nancy Awad, Ashok Natarajan, Badarinath Chickballapur Ramachandrachar and Dr Kavitha Ganesha.

## Author Contributions

**Conceptualization:** Ahmed Hamden Al-Jedai, Hajer Yousef Almudaiheem, Dema Abdulrahman Alissa, Hadi Saeed Al-Enazy, Ghazwa B. Korayem, Ahlam Alghamdi, Shabab Alghamdi.

**Methodology:** Ahmed Hamden Al-Jedai.

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
