## [Decision Letter · Decision Letter 0]

21 Jun 2022

PONE-D-22-02281Cost of cardiovascular diseases and renal complications in people with type 2 diabetes mellitus in the Kingdom of Saudi Arabia: A retrospective analysis of claims databasePLOS ONE

Dear Dr. Al-Jedai,

Thank you for submitting your manuscript to PLOS ONE. After careful consideration, we feel that it has merit but does not fully meet PLOS ONE’s publication criteria as it currently stands. Therefore, we invite you to submit a revised version of the manuscript that addresses the points raised during the review process.

About your paper entitled “Cost of cardiovascular diseases and renal complications in people with type 2 diabetes mellitus in the Kingdom of Saudi Arabia: A retrospective analysis of claims database” We have considered that the paper is interesting and could potentially be published in a new versión (the decisión is Minor Revision) that takes into account the observations made by the referees.

I am attaching the referee's comments, which will help to explain the reasons for our decision. As you can see, the reviewer finds the paper to be of interest, but raises a number of significant concerns. I hope the reports may be useful if you are considering revising the paper for re-submission to Plos One.

We look forward to receiving your revised manuscript.

Kind regards,

Sónia Brito-Costa, Ph.D.

Academic Editor

PLOS ONE

Journal Requirements:

3. Thank you for stating the following in the Acknowledgments/Funding Section of your manuscript: 

The study was funded by NovoNordisk Saudi Arabia

Reviewers' comments:

Reviewer's Responses to Questions

**Comments to the Author**

1. Is the manuscript technically sound, and do the data support the conclusions?

Reviewer #1: Yes

Reviewer #2: Yes

2. Has the statistical analysis been performed appropriately and rigorously? 

Reviewer #1: Yes

Reviewer #2: Yes

3. Have the authors made all data underlying the findings in their manuscript fully available?

Reviewer #1: Yes

Reviewer #2: Yes

4. Is the manuscript presented in an intelligible fashion and written in standard English?

Reviewer #1: Yes

Reviewer #2: Yes

5. Review Comments to the Author

Reviewer #1: The abstract is very clear. There are too many keywords in the abstract, I suggest you to review this point. You can

leave only 5 keywords.

Introduction is very clear with the use of recent bibliographic references.

Results and Methods are clear, but please review Table 4 as it seems that the table is not fully appearing.

Please review the conclusion part of the article.

Thank you.

Reviewer #2: The authors sub,itted a research article in which they assessed the healthcare resource utilization and costs related to cardiovascular and renal complications among patients with T2DM. They designed non-interventional, longitudinal, retrospective, cohort study collected secondary data from three insurance claims databases across Kingdom of Saudi Arabia. The authors found that the healthcare resource utilization and cost gradually increased over the 3-year post-index period compared with the 1-year pre-index period. Finally, they suggested that better management of T2DM to limit the associated

cost burden is need of the hour. The strength of the study is a large sample size, the weakness is a retropsective design. Albeit the findings seem to be impressive, I would like to out forward several issues to discuss.

1. Did authors evaluate a bias of the study related to duration of the T2DM complication treating ? Whether both healthcare resource utilization and cost are regarded to be attributed to severity of the disease ?

2. Was sensitivity analysis performe? Please, give an extensive explanation about it.

3. The conclusive part should compose of a clear and practically useful recommendation.

6. PLOS authors have the option to publish the peer review history of their article (what does this mean?). If published, this will include your full peer review and any attached files.

Reviewer #1: No

Reviewer #2: No

---

## [Author Response · Author response to Decision Letter 0]

4 Aug 2022

REVIEWER 1 COMMENTS

Comment 1: The abstract is very clear. There are too many keywords in the abstract, I suggest you review this point. You can leave only 5 keywords.

Response: Thank you for the comments. 

We have revised and retained only 5 keywords

Type 2 diabetes mellitus; cardiovascular disease; claims database; healthcare resource utilization; chronic renal failure

Comment 2:Introduction is very clear with the use of recent bibliographic references.

Results and Methods are clear, but please review Table 4 as it seems that the table is not fully appearing.

Response: Thank you for the comment

We have reviewed Table 4 and taken care of formatting for complete visibility of the table

comment 3: Please review the conclusion part of the article.

Response: Thank you for the comment.

We have revised the conclusion part as below.

The current analyses shed light on the substantial economic burden in patients with T2DM with comorbid CVD and renal diseases. The study observed a greater frequency of hospitalization in patients with T2DM with associated CVD and renal comorbidities. The findings also suggest a gradual increase in average healthcare resource cost over the 1-year post-index period compared with the 1-year pre-index period. 

Evidence suggests that glucagon-like peptide-1 (GLP-1) agonists are associated with reduced risk of CVS and renal events and sodium-glucose-cotransporter-2 (SGLT2) have improved renal outcomes in patients with T2DM. Therefore, early initiation of these T2DM therapies would reduce long-term cardiovascular and renal outcomes and reduce the associated costs in patients with T2DM.

REVIEWER 2 COMMENTS

The authors submitted a research article in which they assessed the healthcare resource utilization and costs related to cardiovascular and renal complications among patients with T2DM. They designed non-interventional, longitudinal, retrospective, cohort study collected secondary data from three insurance claims databases across Kingdom of Saudi Arabia. The authors found that the healthcare resource utilization and cost gradually increased over the 3-year post-index period compared with the 1-year pre-index period. Finally, they suggested that better management of T2DM to limit the associated cost burden is need of the hour. The strength of the study is a large sample size, the weakness is a retrospective design. Albeit the findings seem to be impressive, I would like to out forward several issues to discuss.

Comment 1: Did authors evaluate a bias of the study related to duration of the T2DM complication treating? Whether both healthcare resource utilization and cost are regarded to be attributed to severity of the disease?

Response: Thank you for the comment

In the current study, HCRU and cost was evaluated in T2DM patients with one or multiple comorbidities (CVD/ renal disease). Claims and costs pertaining to visit and activity types were evaluated for pre-index and post-index period across all three databases. An increase in costs incurred based various visit and activity types was noted during the post-index period as compared to pre-index period. 

Generally, T2DM with CVD or renal complication will be associated with increased cost with severity and duration of disease. However, in the current study, we could not assess the effect of severity of disease on healthcare cost, as the patient number reduced with the follow-up period. We were not able to rule out any bias related to duration and severity of the T2DM complication.

Comment 2: Was sensitivity analysis performed? Please, give an extensive explanation about it.

Response: Thank you for the comment

Sensitivity analysis was not performed in the current study. The study included data from 3 insurance claims databases in KSA covering both the local Saudi population and expatriate population working in the private sector. Each database included longitudinal, anonymized, patient level data, representing a geographically varied range of administrative claims information and capturing data from hospitals, pharmacies, clinics, polyclinics, laboratories, and other providers.

In the current study, descriptive analysis was used to analyse the patient demographics (age, gender and nationality) and HCRU and costs. As the patient numbers in the 1 year post-index period are high and we have used the average cost, which could be representative of true cost.

Comment 3:The conclusive part should compose of a clear and practically useful recommendation.

Response: Thank you for the comment

We have revised the conclusion part as below.

The current analyses shed light on the substantial economic burden in patients with T2DM with comorbid CVD and renal diseases. The study observed a greater frequency of hospitalization in patients with T2DM with associated CVD and renal comorbidities. The findings also suggests a gradual increase in average healthcare resource cost during 1-year post-index period compared with the 1-year pre-index period. 

Evidence suggests that glucagon-like peptide-1 (GLP-1) agonists are associated with reduced risk of CVS and renal events and sodium-glucose-cotransporter-2 (SGLT2) have improved renal outcomes in patients with T2DM. Therefore, early initiation of these T2DM therapies would reduce long-term cardiovascular and renal outcomes and the associated costs in patients with T2DM.

Editors Comments

Comment 1: Please ensure that your manuscript meets PLOS ONE's style requirements, including those for file naming. The PLOS ONE style templates can be found at

Response: Thank you for the comments

Manuscript content has been amended appropriately to meet the PLOS ONE style template requirements pertaining to “Manuscript body formatting”, “Author list and affiliations formatting”, “Supporting Information formatting”

Comment 2:We note that the grant information you provided in the ‘Funding Information’ and ‘Financial Disclosure’ sections do not match. 

Response: Thank you for the comment

During resubmission, appropriate changes to funding information will be made

Comment 3: Thank you for stating the following in the Acknowledgments/Funding Section of your manuscript: 

The study was funded by NovoNordisk Saudi Arabia

Response: Thank you for the comment

As suggested, we have removed the funding related information from the manuscript. 

We have updated the Funding statement as stated below shall include the below amended statement within the cover letter 

The study was funded by Novo Nordisk, KSA and had role in the study design, study methodology and decision to publish.

Comment 4: Please review your reference list to ensure that it is complete and correct. If you have cited papers that have been retracted, please include the rationale for doing so in the manuscript text, or remove these references and replace them with relevant current references. Any changes to the reference list should be mentioned in the rebuttal letter that accompanies your revised manuscript. If you need to cite a retracted article, indicate the article’s retracted status in the References list and also include a citation and full reference for the retraction notice.

Response: Thank you for the comments

We have reviewed the reference list and the list is complete and correct. None of the cited references have been retracted. We have not made any changes to the existing reference list.

---

## [Decision Letter · Decision Letter 1]

17 Aug 2022

Cost of cardiovascular diseases and renal complications in people with type 2 diabetes mellitus in the Kingdom of Saudi Arabia: A retrospective analysis of claims database

PONE-D-22-02281R1

Dear Dr. Al-Jedai,

We’re pleased to inform you that your manuscript has been judged scientifically suitable for publication and will be formally accepted for publication once it meets all outstanding technical requirements.

Kind regards,

Sónia Brito-Costa, Ph.D.

Academic Editor

PLOS ONE

Additional Editor Comments (optional):

Reviewers' comments:

Reviewer's Responses to Questions

**Comments to the Author**

1. If the authors have adequately addressed your comments raised in a previous round of review and you feel that this manuscript is now acceptable for publication, you may indicate that here to bypass the “Comments to the Author” section, enter your conflict of interest statement in the “Confidential to Editor” section, and submit your "Accept" recommendation.

Reviewer #1: All comments have been addressed

Reviewer #2: All comments have been addressed

2. Is the manuscript technically sound, and do the data support the conclusions?

Reviewer #1: Yes

Reviewer #2: Yes

3. Has the statistical analysis been performed appropriately and rigorously? 

Reviewer #1: Yes

Reviewer #2: Yes

4. Have the authors made all data underlying the findings in their manuscript fully available?

Reviewer #1: Yes

Reviewer #2: Yes

5. Is the manuscript presented in an intelligible fashion and written in standard English?

Reviewer #1: Yes

Reviewer #2: Yes

6. Review Comments to the Author

Reviewer #1: The authors submitted an article in which they assessed the healthcare resource utilization and costs related to cardiovascular and renal complications among patients with T2DM. All the questions were answered by the authors.

Reviewer #2: The authors submitted a revised version of the paper along with a comprehensive explanation of the ways by which the corrections were made.

7. PLOS authors have the option to publish the peer review history of their article (what does this mean?). If published, this will include your full peer review and any attached files.

Reviewer #1: No

Reviewer #2: No

---

## [Editor Report · Acceptance letter]

11 Oct 2022

PONE-D-22-02281R1 

Cost of cardiovascular diseases and renal complications in people with type 2 diabetes mellitus in the Kingdom of Saudi Arabia: A retrospective analysis of claims database 

Dear Dr. Al-Jedai:

I'm pleased to inform you that your manuscript has been deemed suitable for publication in PLOS ONE. Congratulations! Your manuscript is now with our production department. 

Kind regards, 

on behalf of

Dr. Sónia Brito-Costa 

Academic Editor

PLOS ONE